# Genome Characterization of Nocturne116, Novel *Lactococcus lactis*-Infecting Phage Isolated from Moth

**DOI:** 10.3390/microorganisms9071540

**Published:** 2021-07-20

**Authors:** Nikita Zrelovs, Andris Dislers, Andris Kazaks

**Affiliations:** Latvian Biomedical Research and Study Centre, Ratsupites 1 k-1, LV-1067 Riga, Latvia; nikita.zrelovs@biomed.lu.lv

**Keywords:** *Lactococcus lactis*, bacteriophage isolation, insect-associated phage, whole genome sequencing, complete genome, genome annotation, comparative genomics, phage diversity

## Abstract

While looking for novel insect-associated phages, a unique siphophage, Nocturne116, was isolated from a deceased local moth specimen along with its host, which was identified by 16S rRNA gene sequencing as a strain of *Lactococcus lactis*. Next-generation sequencing and the subsequent genome annotation elaborated on herein revealed that the genome of Nocturne116 is a 25,554 bp long dsDNA molecule with 10 bp long 3′ cos overhangs and a GC content of 37.99%, comprising 52 predicted open reading frames. The complete nucleotide sequence of phage Nocturne116 genome is dissimilar to any of the already sequenced phages, save for a distant link with *Lactococcus* phage Q54. Functions for only 15/52 of Nocturne116 gene products could be reliably predicted using contemporary comparative genomics approaches, while 22 of its gene products do not yet have any homologous entries in the public biological sequence repositories. Despite the public availability of nearly 350 elucidated *Lactococcus* phage complete genomes as of now, Nocturne116 firmly stands out as a sole representative of novel phage genus.

## 1. Introduction

*Lactococcus* is a genus of gram-positive lactic acid bacteria which have been initially isolated from a variety of different sources (e.g., thermite gut [1], activated sludge foam [2], diseased fish [3,4], and fresh or fermented foods and milk [5,6,7]). The most well-known species of the genus, *Lactococcus lactis*, has undoubtedly been accompanying the “biotechnological” progress of mankind since the dawn of time (although unbeknownst to it for the most of its history), and its strains are still being widely used for the fermentation of milk as a part of starter cultures in the dairy industry [8,9]. Phages are usually omnipresent in any environments where their hosts thrive and virulent lactococcal phage introduction is still being recognized as a threat to successful fermentation and occasionally may lead to product downgrading accompanied by financial losses in the dairy industry despite the technological and strategical advancements for their biocontrol in such environments [10].

Although *Lactococcus* is among the bacterial genera for which the most complete genome sequences of corresponding phages (349 complete genome sequences) are currently publicly available, the genome of Nocturne116 yet again highlights the immense genomic and proteomic diversity of phages, showing that phages described up to date, indeed, represent only the tip of the iceberg of phage diversity.

In this study, we report the isolation, genome sequencing and assembly, as well as genomic characterization of a highly divergent novel *Lactococcus* phage Nocturne116, providing an in-depth elaboration on the rationale behind assigning its open reading frame products with a putative function using comparative genomics approaches employing free and publicly available tools/databases. Additionally, the diversity of all currently known and sequenced *Lactococcus* phages is being investigated in light of the ever-changing phage taxonomy.

## 2. Materials and Methods

### 2.1. Phage and Host Isolation

While searching for novel insect-associated bacteriophages, a dead specimen of local moth species (presumably *Lymantria dispar*) was collected on the living premises and stored at +4 °C overnight.

The specimen was crushed with a pestle and resuspended in 5 mL of physiological saline. After sedimentation of solid particles on the bench of Eppendorf 5424 centrifuge (Eppendorf, Hamburg, Germany) at 2348× *g* for 5 min, 50 µL of supernatant was spread on three Petri dishes with LB agar (g/L: Tryptone—10 (Sigma-Aldrich, St. Louis, MO, USA), Yeast Extract—5 (Sigma-Aldrich), NaCl (Sigma-Aldrich)—10, Bacto Agar (Sigma-Aldrich)—15), and Petri dishes were incubated at room temperature (RT) for two days. Obtained bacterial colonies were purified by subculturing two more times. Subsequently, 5 mL of phage indicator cultures (IC) was prepared in LB liquid media from the single colonies by overnight incubation at RT. After, 50 µL of individual ICs was tested for the absence of spontaneous putative prophage induction and phage contamination by double-agar overlay method (top layer—7 g of agar per L of LB broth) before being used for phage search in the previously obtained supernatant of moth material. To screen the crushed insect suspension for phages, the supernatant was further clarified by centrifugation for 30 min at 8228× *g* on an Eppendorf 5810R centrifuge (Eppendorf) and filtered through a 0.45 μm pore size syringe filter (Sarstedt, Nümbrecht, Germany). Aliquots of filtrate (5, 10, and 50 µL) were then mixed with 50 µL of previously obtained individual indicator cultures and 6 mL of melted soft agar followed by the incubation of several parallel Petri dishes at RT and +30 °C. Phage Nocturne116 was isolated on the lawn of bacterial indicator culture named “LNT” which formed pale semitransparent colonies, and which was identified later to be a strain of *Lactococcus lactis* by Sanger-based sequencing [11] of its 16S rRNA gene PCR product using conventional universal 27F and 1492R primers (ordered at Metabion, Steinkirchen, Germany) [12] and BigDye^®^ Terminator v3.1 Cycle Sequencing Kit (Applied Biosystems, Waltham, MA, USA) according to the manufacturer’s recommendations (Appendix A). Nocturne116 forms semitransparent plaques both at RT and + 30 °C, with larger plaques formed at 30 °C. 

### 2.2. Phage Propagation and Purification

Before propagating the phage, Nocturne116 was plaque-purified by subculturing individual plaques twice as described elsewhere [13] and subsequently propagated using double-agar overlays with 20 Petri dishes to achieve the confluent lysis of the bacterial lawn. Top soft agar layers containing lysed bacterial lawns and phage progeny were collected, vortexed for ~30 s after addition of 5 mL LB media per soft agar from one plate, incubated at RT for ~ 30 min, and then centrifuged at 8228× *g* on an Eppendorf 5810R centrifuge (Eppendorf) for 30 min before carefully decanting supernatant which was subsequently filtered through a 0.45 μm pore size syringe filter (Sarstedt). These steps yielded about 100 mL of the phage-containing filtrate with ~2 × 10^8^ pfu/mL. Purification of the propagated phage was carried out by gel-filtration on 4 Fast Flow Sepharose (Cytiva, Marlborough, MA, USA) followed by Q Sepharose High Performance (Cytiva) ion exchange chromatography. 

Phage presence in peak fractions (PBS buffer of ~7.3 pH as medium) was analyzed by titration and TEM. Finally, the purified phage was concentrated on Amicon Ultra-15, 100 K MWCO filters (Merck, Kenilworth, NJ, USA) to 0.5 mL at 3214× *g* on an Eppendorf 5810R centrifuge (Eppendorf).

### 2.3. Transmission Electron Microscopy and Virion Dimension Measuring

The purified phage specimen (5 μL) was allowed to adsorb on a formvar/carbon-coated copper grid (Sigma-Aldrich) for 5 min and was negatively stained with 0.5% uranyl acetate and allowed to dry for 2 h thereafter before being examined under a JEM-1230 transmission electron microscope (JEOL, Akishima, Japan). Phage particle dimensions represent an average ± standard deviation of a corresponding feature across 10 randomly selected intact phage particles from the best micrograph with numerous particles in the field of view taken with Morada 11 MegaPixel TEM CCD microscope-mounted camera (Olympus, Tokyo, Japan) (Appendix A). Dimensions were measured using ImageJ v1.52a software (straight line for head diameters and segmented line for tail lengths) using a scale bar for a pixel to nm ratio [14].

### 2.4. Phage Genomic DNA Extraction and Whole Genome Sequencing

The purified phage sample (300 µL) was treated with 50 μg of proteinase K (20 mg/mL; ThermoFisher, Wlatham, MA, USA) and with an addition of SDS (Sigma-Aldrich) up to a 0.5% of final concentration before being incubated for an hour at +56 °C and subjected to DNA extraction using Genomic DNA Clean & Concentrator-10 (Zymo Research, Irvine, CA, USA) according to the manufacturer’s guidelines. 

To prepare the input for the NGS library, 200 ng of phage DNA was randomly physically sheared using Covaris S220 focused-ultrasonicator (Covaris, Woburn, MA, USA) with a target fragment length of 550 bp. Fragmented DNA was used as an input for barcoded TruSeq DNA Nano Low Throughput Library Prep Kit (Illumina, San Diego, CA, USA) as per the manufacturer’s reference guide using adapter 7 from TruSeq DNA Single Indexes Set A (Illumina). The quality and quantity of the final library were verified using an Agilent 2100 bioanalyzer (Agilent, Santa Clara, CA, USA) with a high Sensitivity DNA kit (Agilent) and Qubit fluorometer (Invitrogen, Waltham, MA, USA) dsDNA high-sensitivity quantification assay (Invitrogen). Library sequencing was carried out on the Illumina MiSeq system (Illumina) using a 500-cycle MiSeq Reagent Kit v2 nano (Illumina) as one of the 12 pooled differently barcoded libraries.

### 2.5. Genome De Novo Assembly and Validation

Sequencing of the corresponding library yielded 52,851 paired-end reads of up to 250 bp in length. Any of the remaining adapter sequences were removed and quality-trimming of the bases below 20 Phred quality score was performed using the bbduk tool from bbmap package [15], subsequently discarding reads shorter than 50 bp. Processed reads were then used for de novo assembly using SPAdes v. 3.14.0 in isolate mode [11]. PhageTerm v1.0.12 [12] was run on a scaffold representing the complete genome of a novel phage using untrimmed reads. Raw reads were mapped unto the PhageTerm reorganized sequence of a scaffold using BWA-MEM [16] and the sequence alignment map was manually inspected for any coverage dips and assembly ambiguities in Integrative Genomics Viewer [17].

Custom primers (Nocturne116_Fw: 5′-GTTGCAAATAAATGACATTACAC-3′, Nocturne116_Rv:5′-GTGACCTCCTTTGTCTATGC-3′ (ordered at Metabion)) were designed for a Sanger sequencing-based “run-off primer walking” experiment to elucidate the exact phage genome termini. Approximately 96% of the raw reads could be mapped onto the resultant genome providing an average genome coverage of 991x.

### 2.6. Nocturne116 Genome Functional Annotation

Complete genome nucleotide sequence of Nocturne116 was subjected to open reading frame (ORF) prediction using Glimmer [18] and GeneMark [19] and tRNA gene finding with Aragorn [20] and tRNAscan [21] as implemented in DNAmaster v 5.23.6 (https://phagesdb.org/DNAMaster/, accessed on 26 April 2021). Four possible start codons (ATG, GTG, CTG, and TTG) were allowed and only putative ORFs encoding a product of >30 amino acids (aa) in length were considered further. When choosing a putative start codon for any given ORF, initial preference was given to the start codon corresponding to the longest possible ORF product, disallowing ORF overlaps of more than 100 bp.

Corresponding amino acid sequences for the translational product of each ORF was used as a query against (in early March 2021): (1) NCBI conserved domain database (CDD) using NCBI conserved domain search [22] under default settings; (2) against the non-redundant protein sequence database using BLASTp [23] with an e-value threshold of 1 × 10^−3^; (3) HHpred [24] against the Protein Data Bank (PDB), Pfam, UniProt-SwissProt-viral70 and NCBI CD databases under default parameters. Start codon positions were manually evaluated and corrected, where necessary and appropriate, to match those of the homologous entries from the non-redundant protein sequence database, given the sufficient evidence and favorable context in the genome of Nocturne116.

Twenty 5′-3′ bases upstream of the selected putative start codons for each ORF were extracted and inspected for the presence of putative Shine-Dalgarno (SD) sequence complementary to the antiSD sequence of *L. lactis* (13 bases of the *L. lactis* 16s rRNA tail; 3′-UUUCCUCCACUAG-5′ [25]) while allowing for the wobble G-U base pairing. After that, a change in the free energy (kcal/mol), required to bring the two strands of nucleotides (putative SD sequence-containing sequence upstream of each Nocturne116 ORF and *L. lactis* antiSD sequence) together and subsequently form a double helix structure, thus very roughly representing the “likeliness” of translation initiation from a selected start codon, was calculated using free_align.pl script in “helix only” (-o) mode to show the duplex with the lowest possible free energy [26].

### 2.7. Lactococcus Phage Complete Genome Entry Overview

For retrieval of all of the publicly available *Lactococcus* phage complete genomes along with their annotations (*.gb and *.fasta files), the NCBI Nucleotide database was queried to filter submissions available at RefSeq and International Nucleotide Sequence Database Collaboration (INSDC) employing the following search filter: “viruses[filter] AND Lactococcus[Title] AND (phage[Title] OR virus[Title] AND gbdiv phg[PROP]) AND biomol_genomic[PROP] AND (complete genome[Title] OR complete sequence[Title])”. 

Selected metadata associated with genome submissions (accession, title, genome length, CDS count, database entry associated taxonomic information, host) was retrieved using custom python script built around regular expressions and SeqRecord class utility provided within BioPython. Duplicate entries of the same phage found in both RefSeq and INSDC were manually checked and removed, favoring INSDC accessions and leaving 349 non-duplicate entries. The annotated genomes of these entries were next scanned for the presence of CDS features, retrieving phage title, protein_id, product annotation, start codon and aa translation for each feature of type CDS using custom script built around SeqRecord BioPython class. 

The four most common phage start codon (ATG, TTG, GTG, CTG) usage frequencies were then derived (number of CDS starting with a particular start codon divided by the total CDS number that a respective genome entry has) for each of the retrieved phages along with the percentage of unannotated gene products (number of CDS labeled either “hypothetical protein” or “unknown”, or containing “uncharacterized” in their product qualifier, divided by the total CDS number the respective genome entry has).

### 2.8. Lactococcus Phage Comparative Genomics/Proteomics Based Clustering

Intergenomic nucleotide sequence similarities between *Lactococcus* phage genomes that had no genus-level taxonomic information associated with the entry and nine randomly selected representatives from the two *Lactococcus* phage genera (according to the taxonomy associated within the submissions) plus the type isolates for these genera (*Skunavirus*—*Lactococcus* virus sk1 (NC_001835.1), *Ceduovirus*—*Lactococcus* virus c2 (NC_001706.1)) were calculated using VIRIDIC (*n* = 88 + (9 + 1) × 2 = 108) [27], with the genus clustering threshold set at 65%.

Clustering of all publicly available (*n* = 349) *Lactococcus* phages based on their encoded proteome contents was performed using vConTACT2 v0.9.22 [28] under default settings, but without using any non-*Lactococcus* phage sequences. The protein sharing network was visualized in Cytoscape v3.8.2 [29] using edge-weighted spring-embedded layout specifying pairwise similarity scores of phages as weights, so that genomes sharing more protein clusters would be closer to each other [30]. Nodes of the network were colored according to the corresponding phage VIRIDIC cluster assignment if the VIRIDIC cluster was not a singleton (however, corresponding VIRIDIC cluster assignment was extended to all of the *Skunavirus* and *Ceduovirus* phages, while only 10 sequences from both genera were used for VIRIDIC analysis).

## 3. Results and Discussion

### 3.1. Overview of Nocturne116 Genome

Intact virions of Nocturne116 demonstrate a siphovirus morphology having 59.1 ± 1.5 nm × 40.1 ± 1.2 nm prolate capsids with 112.0 ± 8.1 nm long and 9.5 ± 0.8 nm wide non-contractile tails (Appendix A), encompassing a 25,554 bp long dsDNA molecule with 10 bp long 3′ cohesive terminal overhangs (5’-CGCAGTAACT-3’), a GC content of 37.99% and a predicted coding capacity of 94.71% based on the length of its 52 open reading frames (counting overlaps of ORFs once), which corresponds to the complete genome of *Lactococcus* phage Nocturne116 therein. 

BLASTN search querying the complete nucleotide sequence of Nocturne116 revealed striking dissimilarity to other phage genomes, save for an alignment to the genome of *Lactococcus* phage Q54 (DQ490056.1; Query coverage of 36%, 68.08% percent identity, Figure 1), which, as of now, is also the best scoring BLASTN hit and shows ~24.5% similarity (query coverage x identity) to Nocturne116. Sequences from the genomes of different *Lactococcus* species and other *Lactococcus* phage genomes have shown considerably weaker alignments of up to 5% query coverage and up to 76.31% identity to Nocturne116, corresponding to their less than 5% overall similarity. 

It is worth noting that, despite high overall nucleotide sequence divergence to Nocturne116, Q54 [31], the closest relative of Nocturne116 boasts not only similar genome size, organization, and ORF count (26537 bp with predicted 47 ORFs for Q54 versus 25,554 bp with 52 predicted ORFs for Nocturne116), they both also show exactly the same 3′ cohesive genome termini and similar virion morphology (prolate capsids of 56 × 43 nm for Q54 and ~59 × 40 nm for Nocturne116 with 109 nm versus ~112 nm long non-contractile tails, respectively).

**Figure 1 microorganisms-09-01540-f001:**
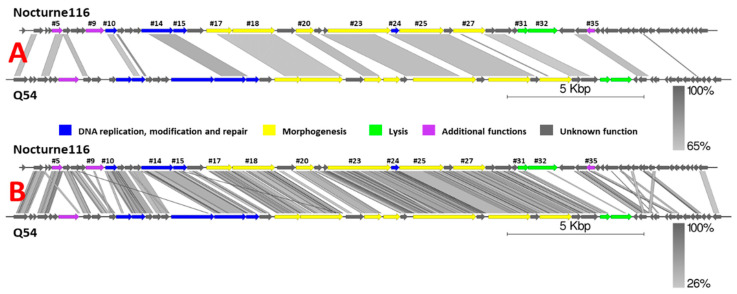
Pairwise genome nucleotide sequence comparison of *Lactococcus* phages Nocturne116 and Q54 using BLASTN (**A**) and TBLASTX (**B**). Genomes are drawn to scale; the scale bar indicates 5000 base pairs. Arrows representing open reading frames point in the direction of transcription and are color-coded based on the function of their putative product according to the legend. Gray boxes represent regions of similarity between genomes and are gradient-colored according to their identity; darker shade of gray represents higher identity. Numbers above selected arrows represent functionally annotated ORFs of Nocturne116 (Appendix A). The pairwise comparisons were carried out using EasyFig [32].

Out of 52 open reading frames predicted in the genome of Nocturne116, ATG was supposed to serve as a start codon 48 times, TTG—twice, while CTG and GTG each had a single occurrence. Based on the minimum free energy calculations for possible helix formation between 20 bases upstream of the predicted start codon and the last 13 bases of *L. lactis* 16s rRNA tail (3’-UUUCCUCCACUAG-5’), 41/52 (78.85%) ORFs were determined to have a Shine-Dalgarno sequence candidate with reasonable binding strength identifiable. No tRNA genes were found in the genome of Nocturne116 (Appendix A).

The proteome of Nocturne116 seemed rather interesting: 22 out of 52 predicted proteins had no homologs available in the public amino acid sequence databases, and another 15 ORF product homologs could be found only in the proteome of *Lactococcus* phage Q54. Overall, 37/52 (~71%) proteins encoded by Nocturne116 remained without even the slightest functional annotation. Functions for only 15 remaining products could be somewhat reliably assigned/predicted using comparative genomics approaches (Appendix A).

Amino acid sequences of four gene products corresponding to evolutionary conserved proteins found in phages (putative DNA polymerase, major capsid protein, terminase, and tail tape measure protein) were chosen as “markers” to inspect the putative evolutionary relationships of Nocturne116 with other publicly available phages, yet again highlighting how distinct Nocturne116 is from other known phages, save for a distant link to phage Q54 (Figure 1 and Figure 2, Appendix A).

### 3.2. Open Reading Frame Product Functional Assignments

While we firmly believe that each annotation of a phage product exclusively in silico is a hypothesis made by the respective annotation authors, and different authors could annotate the same product based on the same degree of evidence for its putative functions differently, there seems to be a considerable shift towards the use of phage “auto-annotation” software instead of a more time-consuming so-called “manual expert-based/supervised annotation”, with cases of rough misannotation not being seldom due to relying solely on automated approaches. Although undeniably feasible in cases where numerous phage genomes or their parts need to be annotated and analyzed in a reasonable amount of time (e.g., metagenome/metavirome studies), we find a solely automated approach to phage genome annotation unacceptable for studies describing a single/few novel phages as such an approach is subject to loss of putatively valuable information. For example, there are some cases when a study detailing the determination of function/functions of a particular product or products for an already submitted and annotated complete phage genome sequence gets published, but the annotation does not get updated with the novel insights about the respective ORF product functions elucidated in vitro. While during rigorous manual annotation of phage product functions, such occurrences are readily identifiable through the relevant literature searches, corresponding functional assignment will be missed by auto-annotation software if not explicitly present in the relevant homolog sequence-associated metadata. However, even the phage papers describing the “manual” annotation process often lack the detailed “decision-making” description for their phage ORF-encoded product functional assignments that would show the rationale behind the annotation proposed by the authors to the phage researcher community. This being said, below we have tried to elaborate on the phage protein functional assignment approach for our rather peculiar phage using mainly three free web-based annotation tools (Conserved Domain Database Search, BLASTP, and HHpred), and most importantly, detailing our decision-making process before assigning products with a function.

#### 3.2.1. ORF #5

ORF #5 in the genome of Nocturne116 encodes for a 132 aa long product with a predicted MW of 15.10 kD and is leniently assumed by us to encode a putative Cas system-associated protein. Although homologs of this protein of identical length were found mostly in proteomes of different *Lactococcus garviae*, *L. lactis* and some unclassified lactococci strains, some of the *Lactococcus*-infecting phages seemed to encode for a similar protein as well (12/135 hits). The most detailed annotations available were found to be a “DUF2800 domain-containing protein” for two similar sequences belonging to *L. garviae*. As of current knowledge, DUF2800 is a member of the CRISPR/Cas system-associated protein Cas4 conserved protein domain superfamily “Cas4_I-A_I-B_I-C_I-D_II-B”, the hit to which was identified when searching for conserved domains in ORF #5 product using CDD search. Although most of our query sequence could be aligned to pfam10926, the coverage spanned only aa positions 251–358 of the 366 aa long conserved domain (CD), which gave a bit score of 34.88 and an E-value of 7.62 × 10^−3^ according to the current domain model’s position-specific scoring matrix (Pssm-ID:402484). As the evidence for functional assignment of this product is rather weak, and to signify its speculative character, we have opted for the inclusion of “putative” in the corresponding product label while choosing not to ignore the possibility of trying to make a functional assignment.

#### 3.2.2. ORF #9

ORF #9 in the genome of Nocturne116 encodes for a 224 aa long product with a predicted MW of 25.38 kD. All the BLASTP phage hits were exclusive to lactococcal phages and some of the best scoring ones were labeled as “Sak” proteins, the functions of which were previously elucidated in *Lactococcus*-infecting phages [38]. Another type of hits having lesser scores was found with various bacteria, the vast majority of which belonged to either enterococci or lactococci, with annotations corresponding to the “DUF1071 domain-containing protein”. 

While it was lucrative to label this product “Sak” protein, the absence of significant similarity was noted in the aa sequence of *Lactococcus* phage ul36 Sak (ORF252; AAM75760.1), for which the function was determined experimentally. Moreover, ul36 Sak (ORF252) has shown a different conserved domain composition to that of Nocturne116 ORF #9. CDD search for ORF #9 revealed only a presence of DUF1071 domain (pfam06378), for which we could not find even a piece of remote functional evidence. The coverage spanned aa positions 6–170 of ORF #9 to positions 3–148 of 152 aa long conserved domain with a bit score of 176.22 and E-value of 2.20 × 10^−56^, showing aa “insertions” in the query (ORF #9) when compared to CD, according to Pssm-ID: 399403. While HHpred search querying ORF #9 did, indeed, reveal a hit to Rad52_Rad22 domain (pfam04098) found in ul36 Sak, the hit to DUF1071 domain (pfam06378) was more convincing. It is worth mentioning, however, that DUF1071 can be found in some of the aforementioned BLASTP phage hits labeled as Sak proteins as well, implying differences in the subjective evaluation of similar degrees of evidence while performing the annotation using comparative genomics approaches. As for ORF #9 of Nocturne116, we could not establish a confident link to experimental functional evidence that would show DUF1071 correspondence with Sak protein functions; thus, we have opted to annotate the ORF #9 of Nocturne116 as “protein of unknown function DUF1071”.

#### 3.2.3. ORF #10

ORF #10 in the genome of Nocturne116 encodes for a 147 aa long product with a predicted MW of 16.29 kD. When querying the amino acid sequence of this protein using BLASTP, an overwhelming amount of significant hits can be documented with homologs found in various species of lactococci, all of which are single-stranded DNA-binding proteins. CDD search reveals a plethora of hits to conserved single-stranded DNA-binding protein clusters, with the best match to ssDNA-binding protein cluster PRK07275 from *Streptococcaceae*. The coverage spanned the complete length of PRK07275 conserved domain, showing two stretches of aa deletions in ORF #10 of Nocturne116, giving a bit score of 207.43 and E-value of 1.01 × 10^−60^, according to Pssm-ID: 180915. Although the previous observations were enough to make a confident function assignment, HHpred search of ORF #10 has also revealed significant hits to crystal structures of ssDNA-binding proteins of various bacteria, thus complementing the evidence to call the product of Nocturne116 ORF #10 a single-stranded DNA-binding protein.

#### 3.2.4. ORF #14

ORF #14 in the genome of Nocturne116 encodes for a 389 aa long product with a predicted MW of 44.11 kD. BLASTP search using aa sequence of Nocturne116 ORF #14 product has revealed hits from a wide array of different organisms. Topped by hits to *Lactococcus* phages (Figure 2A), similar proteins were also found in various other bacteria from orders *Bacillales* and *Lactobacillales*, labeled either as DNA polymerases or metallophosphoesterases with or without the “putative” prefix among entries named “hypothetical proteins”. Interestingly, sequences from bacterial species belonging to the genus *Lactococcus* were not found among the significant hits at all. CDD search did not reveal even a single significant alignment to any of the conserved domains, while HHpred showed hits to C-terminal metallophosphatase domain of archaea DNA polymerase II, small subunit (cd07386), and structures of the corresponding proteins from *Thermococcus* and *Pyrococcus* archaea. 

Although metallo-dependent phosphatase domains are found in a range of different proteins, we have opted to make an assumption that Nocturne116 ORF #14 is a putative DNA polymerase, taking the genomic context and annotations made by the other *Lactococcus* phage genome submission authors into account.

#### 3.2.5. ORF #15

ORF #15 in the genome of Nocturne116 encodes for a 169 aa long product with a predicted MW of 18.90 kD. The best scoring BLASTP hit was to a RuvC endonuclease from *Lactococcus* phage Q54, which was followed by hits to sequences of similar length labeled either “hypothetical protein” or “crossover junction endodeoxyribonuclease RuvC” from different *Streptococcus* species and *Streptococcus* phages. CDD search has, indeed, revealed a presence of crossover junction endodeoxyribonuclease RuvC domain (cd16962), albeit the alignment of the query spanned only the C-terminal (aa 2 to 90) of 153 aa long conserved domain and gave a bit score of 45.52 and E-value of 1.58 × 10^−6^, according to Pssm-ID: 340813. In the meantime, the best hit from the HHpred query was found to be to a RuvC endonuclease structure from *Lactococcus* phage bIL67 (4KTW_A; ref. [39]) followed by close hits to its counterparts from proteomes of *E. coli*, *Thermus thermophilus*, and *Pseudomonas aeruginosa*. This evidence was deemed to be convincing enough to assign the product of ORF #15 from Nocturne116 with a Holliday junction resolvase function.

#### 3.2.6. ORF #17

ORF #17 in the genome of Nocturne116 encodes for a 314 aa long product with a predicted MW of 35.08 kD. The sole BLASTP hit was to a putative portal protein from *Lactococcus* phage Q54 (YP_762589.1). CDD search did not reveal any conserved domains and the HHpred hits, although topped by ten entries corresponding to portal proteins either with or without “probable” prefix, were chosen not to be trusted on the basis of their probabilities (less than or equal to 95.25%) and E-values (more than or equal to 1). A PSI-blast among sequences of viral origin (taxid: 10239) was employed in this case, resulting in hits to “minor structural”, “capsid and scaffold”, as well as more “portal” proteins, either labeled with or without the “putative” prefix. Apart from the aforementioned hit to *Lactococcus* phage Q54, the other alignments were rather ambiguous and have shown up in 61% of query coverage with 24.34% identity for “capsid and scaffold protein” of *Lactococcus* phage CHPC1242 (QGT52752.1) and ≤ 49% coverage of ≤ 23.68% identity for other hits. It was noted that labeling of these somewhat similar products was inconsistent, although they aligned to the same spot of Nocturne116 ORF #17 product. However, the paper of Fortier and colleagues describing the genome and proteome of phage Q54, the closest known relative of Nocturne116, had employed a mass-spectrometry analysis of structural proteins and revealed that homolog of ORF #17 in Q54 (YP_762589.1) is a structural protein and unlikely to be a major capsid protein (MCP) based on its quantity in the SDS-gel presented therein. In addition, MCP was identified to be encoded by another ORF in the genome of Q54 [31]. These findings, along with the genomic location of Nocturne116 ORF #17 preceding the better MCP candidate by far, have prompted us to predict the portal protein function for its product due to the lack of a better ORF candidate, as the presence of a portal protein is essential for the replication of tailed bacteriophages.

#### 3.2.7. ORF #18

ORF #18 in the genome of Nocturne116 encodes for a 516 aa long product with a predicted MW of 57.50 kD. The highest scoring BLASTP hit was to major capsid protein (MCP) of phage Q54 (YP_762590.1; 99% query coverage and 64.35% identity), while all the following hits were to proteins of varying length, mostly from either *Lactococcus* or *Streptococcus* phages (Figure 2B) and some prophages/prophage remnants of *Listeria innocua*, *L. monocytogenes* and *Streptococcus pneumoniae*, often annotated as major capsid proteins. These “follow-up” hits were notably more different (up to 98% query coverage, although only of 25.37% identity to a second-best hit—MCP of *Lactococcus* phage GE1 (YP_009226660.1), and the majority of the alignments were only to the C-terminal part of the queried protein, beginning roughly at aa position 200 onwards). CDD search revealed an HK97 family phage major capsid protein conserved domain (TIGR01554)—a member of the phage capsid domain superfamily (cl27082). Alignment of ORF #18 aa sequence to TIGR01554 spanned aa positions 145–427 corresponding to positions 31–295 of 386 long conserved domain, showing aa stretch “insertions” in both the query and the conserved domain, and gave a bit score of 54.27 and E-value of 8.96 × 10^−8^, according to Pssm-ID: 273690. As expected, HHpred did reveal convincing hits to MCP UniProt entries and even to determined capsid structures from other phages. It was also noted that some hits have aligned just to an N or C terminus of Nocturne116 ORF #18 product, with distinct prohead protease hits aligning to N terminus, while MCP hits aligned to C terminus. The best HHpred hit, however, did align to a whole length of our query and was aligned to a capsid polyprotein from *Pseudomonas* phage PAJU2 (P85500), for which it was proposed that MCP protein, indeed, might be fused to a prohead protease and autocatalytically cleaved [40]. Combining these observations, we believe that the presence of prohead protease function for ORF #18 cannot be ruled out; thus, the ORF #18 of Nocturne116 might present yet another occurrence of natural fusion between prohead protease and MCP. However, due to the lack of in vitro experimental evidence for Nocturne116 ORF #18 function, we have decided to view this particular ORF as one of the peculiar major capsid protein encoding cases; therefore, we have labeled the Nocturne116 ORF #18 product as MCP in the annotated genome entry (MW791312.1) while detailing our extended finding herein.

#### 3.2.8. ORF #20

ORF #20 in the genome of Nocturne116 encodes for a 220 aa long product with a predicted MW of 23.73 kD. All the BLASTP hits were revealed to be exclusively to *Lactococcus* phage proteins of 205–212 aa length. The best scoring hit was to a major tail shaft protein of phage Q54 (YP_762592.1; 93% query coverage of 67.48% identity), with other hits, mostly labeled as putative tail fiber proteins, being less similar (67–93% query coverage of up to 33.76% identity). CDD search did not reveal any conserved domains, while the top two HHpred search hits (both having E-values ≤ 1.5 × 10^−31^) were to the protein family PF06488 (*Siphoviridae* phage tail tube proteins including several proteins from *Lactococcus lactis*) and UniProt entry of tail tube proteins from *Lactococcus* phage SK1 (O21879). As no substantial evidence to view the product of Nocturne116 ORF #20 as a tail fiber protein was present, “major tail shaft protein” was chosen for product annotation as a synonym of “tail tube protein”, ensuring correspondence to product labels of similar proteins from *Lactococcus* phages Q54 and GE1 (YP_762592.1 and YP_009226662.1, respectively).

#### 3.2.9. ORF #23 

ORF #23 in the genome of Nocturne116 encodes for a 765 aa long product with a predicted MW of 81.45 kD. BLASTP search revealed a 99% query coverage of 63.54% identity hit to the tail tape measure protein (TMP) of phage Q54 (YP_762595.1) followed by numerous hits to proteins from other *Lactococcus*-infecting phages (mainly hits from *Ceduovirus* genus; Figure 2D), all of whom seemed to align only to about first 500 aa from the N terminus of Nocturne116 ORF #23 product, giving query coverage of up to 64% and 38.19% identity—these were labeled as either TMP, “tail assembly”, “tail adsorption”, or “tail/minor tail” proteins. CDD search revealed alignment to a cluster of orthologous proteins COG3941, comprising phage tail tape-measure proteins. Alignment of the query spanned aa 86–413 of ORF #23 product to positions 4–344 of 633 aa long domain, showing aa stretch “insertions” for both the query and the hit, giving a bit score of 49.11 and E-value of 9.12 × 10^−6^, according to Pssm-ID: 226450. 

It was previously noted by other researchers that TMP is usually encoded by the longest ORF in siphophage or myophage genomes and that length of TMP protein roughly corresponds to the tail length of a phage [41,42], which explains the observed protein aa length differences among the probable tape measure protein hits to Nocturne116 ORF #23 when employing HHpred search. In addition to the aforementioned evidence, taking into account that ORF #23 is also the longest open reading frame in the genome of Nocturne116, the tape measure protein function was assigned to its product.

#### 3.2.10. ORF #24

ORF #24 in the genome of Nocturne116 encodes for a 99 aa long product with a predicted MW of 11.80 kD. BLASTP search using Nocturne116 ORF #24 product as a query revealed hits from many *Lactococcus* phages (mainly belonging *Ceduovirus* genera) and a set of similar proteins from a taxonomically diverse range of bacteria representing *Bacteroidetes*, *Firmicutes*, and *Proteobacteria* phyla, implying immense evolutionary conservatism for ORF #24 product. CDD search revealed an alignment to the HNHc nuclease domain (smart00507). Alignment spanned aa positions 30–79 of ORF #23 product to positions 13–52 of 52 aa long HNHc domain (smart00507), showing aa stretch “insertion” in the query and resulting in a bit score of 35.13 and E-value of 5.26 × 10^−4^ according to multi-domain Pssm-ID: 214702. The most probable HHpred hit (98.65% probability) represented an HNH endonuclease structure from *Geobacillus* virus E2 (5H0M_A; Target length 130; E-value of 7.2 × 10^−8^) followed by a hit to UniProt entry of NinG protein of phage Lambda (PF05766.14; Target length 197; E-value of 1 × 10^−7^). Although two of the BLASTP-determined most similar proteins from the proteomes of phages Q54 and GE1 were labeled as “hypothetical” (YP_762596.1; 100% query coverage of 65.05% identity and YP_009226667.1; 96% query coverage of 47.42% identity, respectively), some of the less similar sequences from a few other phage hits, as well as the majority of bacterial hit products, were annotated as “HNH endonuclease”. Thus, the product of Nocturne116 ORF #24 was assigned an HNH endonuclease function.

#### 3.2.11. ORF #25

ORF #25 open reading frame in the genome of Nocturne116 encodes for a 542 aa long product with a predicted MW of 62.44 kD. After querying Nocturne116 ORF #25 product aa sequence, BLASTP hits comprised *Lactococcus*-infecting phage proteins (Figure 2C; again, mainly from the *Ceduovirus* genera, as with some of the preceding ORF products) and similar proteins found in genomes of bacteria from phylum *Firmicutes*, labeled either “terminase” or “terminase large subunit”, with or without “putative” prefix. The best scoring hit was to a terminase protein from phage Q54 (YP_762597.1) which had a query coverage of 94% with an identity of 75.69%, while other hits showed up to 97% query coverage with up to 38.66% identity. It was noted that homologous proteins from the majority of the phages from genus *Ceduovirus* were of identical length (518 aa) and showed identical or near-identical alignment scores (300–303), query coverages (95%), and identity percentages (35.63–36.21%). CDD search has shown that Terminase_1 domain (pfam03354) is present in the aa sequence of Nocturne116 ORF #25 product, with an alignment spanning query positions 71–525 to positions 2–452 of 466 aa long conserved domain, giving a bit score of 188.65 and an E-value of 1 × 10^−53^ according to Pssm-ID: 281364. As was expected, HHpred revealed convincing hits to phage terminase large subunits or terminase UniProt entries as well. The terminase is responsible for phage genetic material packaging into procapsids, thus being an essential protein for tailed phages [43]. 

Although two ORFs encoding, respectively, terminase small (TerS) and large (TerL) subunits, are frequently found in phage genomes, often being in proximity of each other, we have failed to reliably identify a potential TerS-encoding ORF candidate in the genome of Nocturne116 as TerS sequences are not as evolutionarily conserved as their large subunit counterparts. Taking the aforementioned evidence into account, as is usual for cases where no two terminase subunits coding genes can be identified in a phage genome, the product of Nocturne 116 OF #25 was labeled just as a “terminase”.

#### 3.2.12. ORF #27

ORF #27 in the genome of Nocturne116 encodes for a 392 aa long product with a predicted MW of 42.71 kD. CDD and HHpred searches did not reveal any hits and the sole BLASTP hit was found to be a minor structural protein of *Lactococcus* phage Q54 (YP_762599.1; query coverage 100%, identity 51.02%, length 391). It is worth noting, however, that this ~43 kD homolog encoded by ORF30 of phage Q54 (YP_762599.1) was originally annotated based on tandem mass spectrometry analysis and could be seen on SDS-PAGE; thus, we believe that the product of Nocturne116 ORF #27 can be reliably annotated as “minor structural protein” as well, owing to the previous in vitro elucidation of phage Q54 structural proteome by Fortier, Bransi, and Moineau [31].

#### 3.2.13. ORF #31

ORF #31 in the genome of Nocturne116 encodes for a 130 aa long product with a predicted MW of 14.35 kD. HHpred and CDD searches did not reveal any hits, while yet again the sole BLASTP hit was to a similar protein from proteome of Q54—its holin (YP_762602.1; query coverage 94%, identity 45.53%, length 125 aa). TMHMM Server v.2.0 (http://www.cbs.dtu.dk/services/TMHMM/; accessed on 29 April 2021) was used to check the presence of transmembrane helices in the aa sequence of Nocturne116 ORF #31 product—a characteristic signature of holins. The search revealed the presence of three transmembrane helices at ORF #31 product aa 13–32, 47–69, and 76–95, similar to the findings in the aa sequence of Q54 holin by Fortier and colleagues [31]. Therefore, the product of Nocturne116 ORF #31 was subsequently annotated as holin, more specifically, as holins are classified based on the number of transmembrane helices, belonging to class I holins [44].

#### 3.2.14. ORF #32

ORF #32 in the genome of Nocturne116 encodes for a 355 aa long product with a predicted MW of 38.59 kD. Although the majority of the most high-scoring hits were found to a LysM peptidoglycan-binding domain containing proteins from various *Lactococcus* species, confident hits to several *Lactococcus* phages were also present, with the highest scoring phage-derived hit being to an endolysin from *Lactococcus* phage 62,502 (ANT43648.1; query coverage 100%, identity 51.83%, length 429 aa). CDD search revealed alignments to GH25_muramidase (cl10448; glycosyl hydrolase family 25 (GH25) catalytic domain-containing muramidases; positions 1–196/196 of CD aligned to positions 29–222 of the query; bit score of 227.28 and E-value of 5.45 × 10^−74^, according to Pssm-ID: 415848) and LysM (cd00118; lysin motif; positions 1–45/45 of CD aligned to positions 257–301 of the query; bit score of 69.05 and E-value of 2.35 × 10^−15^, according to Pssm-ID: 212,030 [Multi-domain]) domains, as well as short alignment to mltD (cl32574; membrane-bound lytic murein transglycosylase D; 345–438/456 of CD aligned to positions 258–346 of the query; bit score of 68.22 and E-value of 1.32 × 10^−12^, according to Pssm-ID: 182,727 [Multi-domain]) domain. HHpred search results have shown various structures of different phage lysins among the top hits as well. All of the aforementioned evidence, as well as its genomic location next to a predicted holin, suggests that the product of ORF #32 from the Nocturne116 genome is an endolysin.

#### 3.2.15. ORF #35

ORF #35 in the genome of Nocturne116 encodes for a 103 aa long product with a predicted MW of 12.46 kD. BLASTP search revealed numerous hits to either hypothetical proteins or DUF3310 domain-containing proteins from different *Lactococcus* phages (belonging to two of the most-populated currently recognized *Lactococcus* phage genera, with hits belonging to *Skunavirus* being more prominent than those of *Ceduovirus*) and some bacteria. The majority of the hits showed similar query coverages in the range of 88–98% with identities in the range of 29.41–40.85%, and although hit protein lengths varied from 72 to 163 aa, most of them were within 101–116 aa length range. CDD search has indicated the presence of a domain of unknown function 3310 (DUF3310; pfam11753) in the query. Alignment spanned positions 15–45 of the query to positions 15–45 of 60 aa long conserved domain, giving a bit score of 34.99 and an E-value of 5.77 × 10^−4^, according to Pssm-ID: 403068. While it was noted in Pfam entry that DUF3310 represents a family of conserved bacteriophage proteins, their function yet remains to be elucidated. However, the HHpred search top hit was to a UniProt entry of SaV protein from *Lactococcus* phage SK1 (O21894; [45]), with a close second being to the aforementioned DUF3310 present also in CDD search. SaV protein of SK1, indeed, had an identifiable conserved DUF3310 present in its aa sequence as well, presenting a better alignment of CD to SK1 query (7–64 aa positions of SK1 SaV (O21894) aligned to 5–59 aa positions of 60 aa long DUF3310, showing bit score of 40.38 and an E-value of 8.61 × 10^−6^, according to the Pssm-ID: 403068) than the Nocturne116 ORF #35 product query. 

Haaber and colleagues have previously demonstrated that SaV is an early-transcription phage protein that is involved in the sensitivity of some lactococcal phages to the then recently described abortive infection mechanism AbiV, and that SaV shows a toxic effect in *L. lactis* and *E. coli*; thus, it was suggested by the authors of that study that the function of SaV during the phage infection is to shut down or redirect essential host functions [45,46]. While it may seem that DUF3310 might, indeed, be a characteristic of *Lactococcus*-infecting phage SaV-like proteins (although present in proteins from phages of other hosts as well), in ORF #35 of Nocturne116, only a half of the DUF3310 domain length seems to be conserved and, although already noted by Haaber and colleagues that the middle regions of SaV proteins are conserved even in distantly related phages [45] (which is yet again highlighted here hypothetically), we have decided to stray away from SaV protein annotation for the product of Nocturne116 ORF #35 and went for a “protein of unknown function DUF3310” annotation for its product.

### 3.3. The Current State of Lactococcus Phage Genomic Diversity

At the time of writing, there were 349 complete *Lactococcus*-infecting phage genome entries publicly available at the RefSeq and INSDC, which makes *Lactoccocus* one of the most popular host genera for phage isolation. These entries span in length from 18,762 (*Lactococcus* phage asccphi28; EU438902.1) to 132,949 (*Lactococcus* phage AM4; KY554771.1) base pairs, and, respectively, encode from 28 to 194 products. Deduplication of these 349 complete genome entries using CD-HIT v4.8.1 at 0.95 nucleotide sequence identity threshold (current ICTV BAVS phage species demarcation criterion) resulted in 294 putative phage species, more than a half of which are officially recognized by the ICTV.

It was noted that genera-level taxonomical information associated with these entries listed only two genera of *Lactococcus*-infecting phages, with genus *Skunavirus* comprising 214 *Lactococcus*-infecting phage genomes and 47 for genus *Ceduovirus*. Respectively, this made 88 *Lactococcus*-infecting phages still unclassified at the genus level, and we believed that hardly would there be no additional genomic similarity clusters corresponding to additional putative genera among these 88 phages. 

The bacteriophage taxonomy is known to be in rapid flux since the affordability of sequencing has grown and the unraveling of phage diversity began happening at unprecedented rates worldwide. Thus, ICTV’s Master Species List 2020.v1 ratified in March, 2021 and published online on 18 May 2021 was consulted for the most up-to-date official taxonomy/classification of *Lactococcus* phages. The official taxonomy was found to list 158 species and, to our surprise, these 158 species were already scattered among 14 genera in contrast to 2, as seen in the phage genome entry associated metadata, which shows that taxonomical information seen in the public sequence repositories strongly lags behind the officially recognized phage taxonomy overseen by ICTV BAVS. Out of these 14 genera, 7 (*Chertseyvirus*, *Chopinvirus*, *Negarvirus*, *Questintvirus*, *Sandindevirus*, *Uwajimavirus*, *Whiteheadvirus*) were found to contain just a single species each. While *Audreyjarvisvirus* had 7 species listed in the viral species master list, *Vedamuthuvirus* and *Teubervirus* both had 5, *Nevevirus* had 4, and *Fremauxvirus* had 2 species. *Skunavirus* and *Ceduovirus* genera had 94 and 34 species in the official taxonomy, respectively.

Based on the annotation of the Nocturne116 complete genome sequence and its genome encoded protein similarity values to those of other phages, it became obvious that Nocturne116 would be a singleton on the genus level, although having a fair amount of proteome homologous to its distant relative, phage Q54 (a single representative of *Questintvirus* genus). However, its characterization has prompted us to have a look at the overall context of the currently available lactococcal phage genome diversity, with special attention to *Lactococcus* phage genomes yet “unclassified” at the genus level based on the metadata from respective genome sequences. 

This being said, intergenomic similarities between yet “unclassified” *Lactococcus* phage genomes and nine randomly selected representatives from the two *Lactococcus* phage genera listed within the complete genome entries in public sequence repositories, along with the respective type species (*Skunavirus*—*Lactococcus* phage sk1, *Ceduovirus*—*Lactococcus* phage c2), were carried out using VIRIDIC (*n* = 108), while proteome comparisons of *Lactococcus*-infecting phages were done employing the vConTACT2 pipeline and including all the *Lactococcus*-infecting phage genomes available at the time (*n* = 349). Both analyses revealed largely coinciding clusters of phages representing novel recently recognized *Lactococcus*-infecting phage genera. However, as was expected due to the redundancy of genetic code, clustering based on the proteome was more sensitive and able to find evolutionary links between phages very distantly related based on the genome nucleotide sequence alone (Figure 3, Appendix A).

It was also noted that under default settings VIRIDIC did not cluster 9 + 1 selected genomes of *Skunavirus* phages (based on genome metadata-associated taxonomic information) in a single cluster at the ICTV BAVS set threshold for phage assignment into the existing genera based on nucleotide sequence similarity (70% sequence similarity [27]), so the 65% nucleotide sequence similarity threshold was employed to respect the *Skunavirus* genus level taxonomy as seen in the respective genome submissions. With this clustering threshold taken into account, none of the “unclassified” viruses (*n* = 88) were found to belong to either of the two *Lactococcus* phage genera currently recognized in the taxonomical metadata of the submissions, while overall forming 36 clusters (two clusters corresponding to *Skunavirus* (VIRIDIC cluster 5) and *Ceduovirus* (VIRIDIC cluster 26) genera excluded). Only 16 of the nucleotide sequence similarity clusters consisted of two or more phages, with the other 20 “clusters”, including the genome of Nocturne116, being singletons based on the pairwise genome nucleotide sequence similarity demarcation/clustering at 65% (Appendix A). The most populated cluster designated VIRIDIC cluster “14” was found to contain 14 out of the 88 “unclassified” lactococcal phage genomes, corresponding to ICTV recognized genus *Audreyjarvisvirus*; VIRIDIC cluster “3” had nine representatives corresponding to genus *Teubervirus*; VIRIDIC cluster “16”—six phages from genus *Vedamuthuvirus*; VIRIDIC cluster “21”—5 phages yet without standing in the official ICTV taxonomy. The remaining 12 VIRIDIC clusters that were not singletons had between 2 to 4 corresponding phages, respectively (VIRIDIC cluster 17—genera *Nevevirus*, VIRIDIC cluster 11—genera *Fremauxvirus*).

Based on the vConTACT2 clustering of proteomes derived from 349 *Lactococcus* phages, 252 were found to belong to one of the 17 distinct vConTACT (VC) clusters, 78 of the phages showed overlaps between two VC clusters, while proteomes of nine phages overlapped between three or more VC clusters, eight phages were deemed “outliers”, and two of the phages formed singletons. It was noted that all 47 of the phages belonging to genus *Ceduovirus* were clustered into a single proteome cluster (VC 14_0), while the situation with skunaviruses was more ambiguous. Out of 214 phages belonging to genus *Skunavirus*, only 127 were found to be clustered within one of the four proteome clusters (VC1_0—68/127, VC10_0—34/127, VC5_0—22/127, and VC13_0—3/127, respectively), while the majority of the remaining 87 phages identified as belonging to “*Skunavirus*” genera based on the taxonomy associated with the genome submissions have shown considerable overlaps between two or more proteome clusters, and three of them were “outliers”.

These observations might be either suggestive that the time has come to revise the definition of *Skunavirus* genus (at least as was seen from the sequence-associated taxonomy available at INSDC and Refseq at the time of writing), or that the usage possibilities of NCBI taxonomy database [49] are very limited for phages, as even nine randomly selected presumably *Skunavirus* phages failed to make a single cluster around the type species reference strain based on mechanistic 70% nucleotide sequence similarity genus demarcation criterion (as seen from VIRIDIC-generated heatmap [27], Appendix A). The nonhomogeneity of (presumably) *Skunavirus* representatives becomes also evident when comparing the proteome contents of corresponding phages, showing not only well-defined proteome clusters, but a number of phages overlapping their boundaries as well, in contrast to *Ceduovirus*, which holds coherent by both nucleotide sequence similarity and proteome content comparisons (Figure 3).

Combining the results of these comparative analyses, the necessity of the latest changes in the *Lactococcus* phage taxonomy recently ratified by ICTV and not yet taken into the account by public biological sequence repositories becomes evident. At least two novel and comparatively populated genera of *Lactococcus*-infecting phages distinct from other *Lactococcus* phages were recently defined (corresponding to “Viridic_cluster 14 = VC_25_0 ~ ICTV *Audreyjarvisvirus*” with 14 representatives and “Viridic_cluster 3 = VC_26_0 ~ ICTV *Teubervirus*” with 9 representatives, as seen herein). Other sets of phages corresponding to a less populated novel genera are also easily identifiable from these results (e.g., Viridic_cluster 16 = VC_cluster 15_0 ~ ICTV *Vedamuthuvirus* (*n* = 6), Viridic_cluster 17 = VC_cluster 17_0 ~ ICTV *Nevevirus* (*n* = 4), Viridic_cluster 11 = VC_cluster 16_0 ~ ICTV *Fremauxvirus*, etc.) as well (Figure 4).

Some other evolutionary-interconnected, but, nonetheless, distinct enough clusters of *Lactococcus* phages presented herein (Figure 3 and Figure 4) suggest that the taxonomy of yet unclassified *Lactococcus* phages might be worthy of additional attention from ICTV BAVS, while the definition of these putative new genera based on their genomic contents could, however, potentially be hampered by the great variation in the quality of the genome annotation efforts for similar phages by different research groups that does not seem to be related to the genome sizes of phages being annotated (Appendix A, Appendix A). Although the temporal differences (scientific knowledge available on the year of annotation and submission of each particular phage) might explain the variation in fractions of unannotated gene products for any given group of related phages to a degree, with older submissions potentially having less of their ORFs functionally annotated, the phage genome functional annotation process should not be taken lightly.

As an additional note, while making revisions to the initial version of this manuscript during the review process, it was found by us that on 15.07.2021, the genus level taxonomy associated with the *Lactococcus* phage complete genome submissions had finally been updated in public biological sequence repositories to conform with the current official ICTV taxonomy ratified in March 2021 after a lag of a few months.

## 4. Conclusions

While in recent decades, the phage genome annotation process has shifted largely to in silico, often omitting in vitro experiments entirely, where feasible, it should not rely solely on unsupervised auto-annotation software to ensure the most utility of a given annotated genome to the phage researcher community, especially in light of the possibilities that comparative genomic approaches open for phage diversity research. 

Here, we have tried to describe phage−host pair isolation and identification from insect material with a large focus on the isolated *Lactococcus* phage Nocturne116 genome characterization. A detailed description of the Nocturne116 genome annotation process using freely available tools, as well as elaboration on the rationale behind assigning its ORF-encoded products with a function, is given for a phage that is currently without any close relatives, which made it an uneasy annotation target even employing a manual supervised annotation approach.

Despite genus *Lactococcus* being one of the most popular host genera for cultured phages completely sequenced up to date with nearly 350 phage genome entries, *Lactococcus* phage Nocturne116 might represent a novel phage genus, being its sole representative as of now.

## Figures and Tables

**Figure 2 microorganisms-09-01540-f002:**
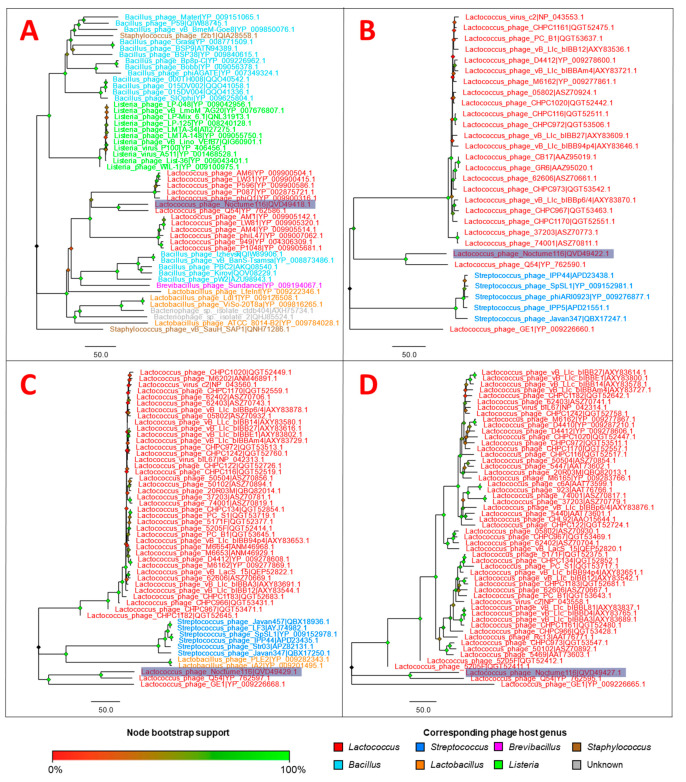
Neighbor-joining (NJ) trees of the selected Nocturne116 protein amino acid sequences and similar sequences found in the proteomes of other phages. The trees are drawn to scale and branch lengths correspond to the number of amino acid differences. For each ClustalW-generated [33] multiple sequence alignment used to build the NJ trees [34], all positions with less than 90% site coverage were eliminated (less than 10% alignment gaps, missing data, and ambiguous bases were allowed at any position). (**A**) putative DNA polymerase (51 sequences with 393 aa positions in the final dataset; the optimal tree with the sum of branch lengths = 2245.23899841 is shown); (**B**) major capsid protein (30 sequences with 394 aa positions in the final dataset; the optimal tree with the sum of branch lengths = 836.47900391 is shown); (**C**) terminase (51 sequences with 507 aa positions in the final dataset; the optimal tree with the sum of branch lengths = 1099.23297322 is shown); (**D**) tail tape measure (54 sequences with 619 aa positions in the final dataset; the optimal tree with the sum of branch lengths = 11,555.21886359 is shown). In all of the trees: tips are labeled as “Phage|respective protein accession” and labels are colored based on the host genera of a phage from the proteome of which the respective sequence was retrieved according to the legend; label of the leaf containing the amino acid sequence of Nocturne116 is highlighted in blue; nodes are colored based on the percentage of their bootstrap support (inferred from 1000 bootstrap test replicates [35]) in gradient from red (low) to green (high) according to the legend. Evolutionary analyses were conducted in MEGA7 [36] and the resulting trees were drawn in FigTree v1.4.4 [37].

**Figure 3 microorganisms-09-01540-f003:**
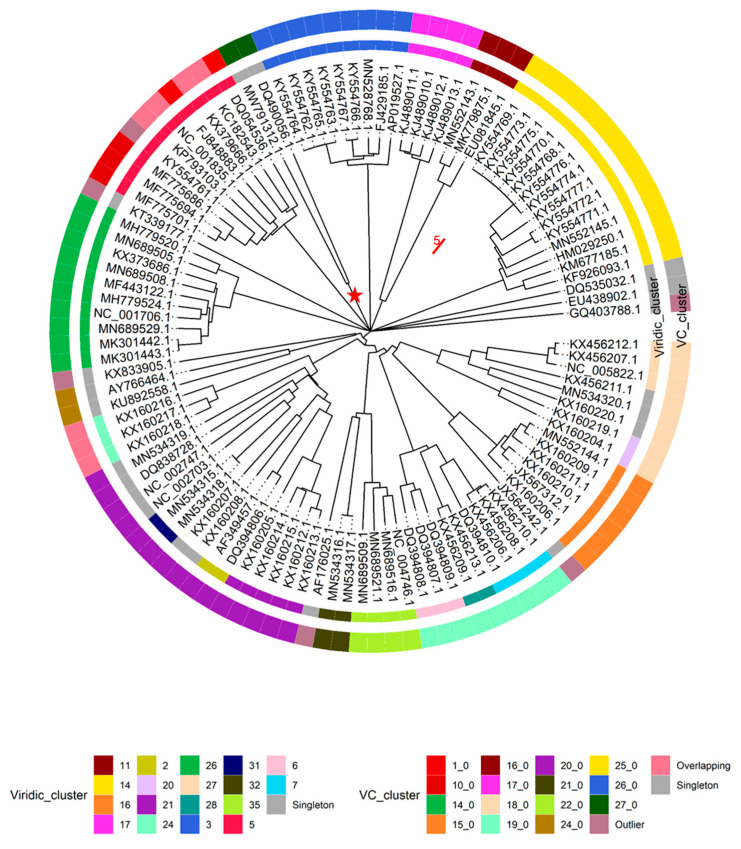
Genome nucleotide sequence similarity (VIRIDIC) and proteome content similarity (vConTACT2) based clustering comparison of *Lactococcus* phages without genus level taxonomical information in complete genome entry associated metadata. The NJ circular tree was drawn to scale from a pairwise nucleotide sequence distance matrix generated by VIRIDIC (*n* = 108); scale bar represents 5% nucleotide sequence divergence (phage species level complete genome nucleotide sequence similarity demarcation criterion). Tips are labeled according to genome accessions. The inner ring shows the results of VIRIDIC clustering at a 65% sequence similarity threshold and is colored according to the “Viridic_cluster” legend (*n* = 108 analyzed and shown). The outer ring shows the results of proteome similarity-based clustering of *Lactococcus* phages employing vConTACT2 and is colored according to the “VC_cluster” legend (*n* = 348 analyzed, *n* = 108 shown). Ten phages corresponding to “Viridic_cluster 5” (red inner ring part) and “Viridic_cluster 26” (green inner ring) are 9 randomly selected +1 entries that were classified as, respectively, *Skunavirus* or *Ceduovirus* in complete genome entry associated metadata. The branch leading to “MRCA” node of phages Nocturne116 and Q54 is annotated with a red star. The tree was created using “ape” [47] and annotations were visualized using “ggtree” [48] R packages.

**Figure 4 microorganisms-09-01540-f004:**
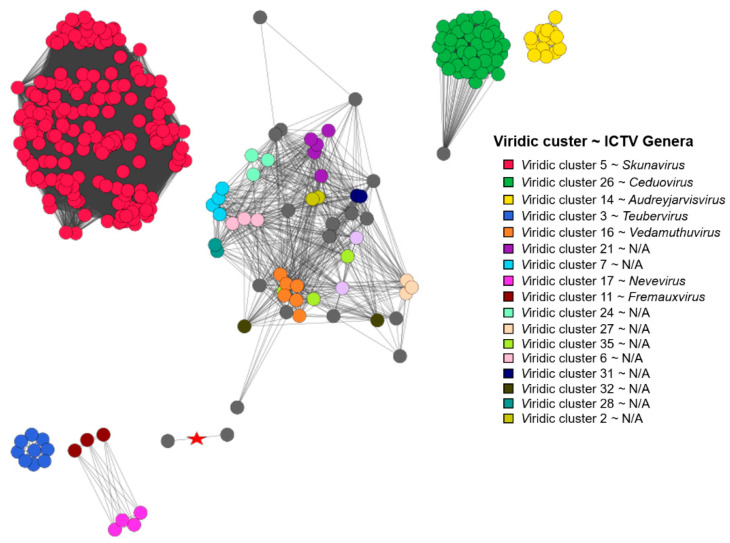
vConTACT2 protein sharing network of 347 *Lactococcus* phages (phages KSY1 (DQ535032.1) and asccphi28 (EU438902.1) were determined to be singletons based on their proteome contents and are not depicted). Nodes are colored based on the VIRIDIC cluster assignment (corresponding to the 65% genome nucleotide sequence similarity) according to the legend. The edge connecting Nocturne116 and Q54 is annotated with a red star. Of note, as VIRIDIC cluster assignment was not performed for all the *Skunavirus* and *Ceduovirus* representatives, the respective red and green coloring was extended on the basis of taxonomical assignment seen in the respective genome entries.

## Data Availability

Annotated complete genome sequence of *Lactococcus* phage Nocturne116 reported herein is available at GenBank under accession MW791312.1. Accession numbers of other phages used in the study are listed in Appendix A.

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
