# Peer review of "Genome Characterization of Nocturne116, Novel Lactococcus lactis-Infecting Phage Isolated from Moth"

_microorganisms, 2021, doi:10.3390/microorganisms9071540_

Round 1

Reviewer 1 Report

General comment

It is a well done work that deserves to be published by this journal after the below minor issues have been rectified.

Parts that needs revision are:

Materials and Method Section

Please use RCF units (…xg) instead of RPM units

State the product catalogue number company name; for all the material (e.g media, enzymes and filters) used. (L100, L103, L106, L109-110, L125 )

As for machines like TEM: state their model/name, include the town or city and country where the equipment was made (L92, L95, L105, L112).

 L85:     …QHP IEX- State its full meaning for the readers.

L86-87:

  1. Which medium was used during phage concentration? ...and at what RCF (..xg) were samples spun?
  2. Alternatively you may cite the authors whose method was used.

L142: State an e-value threshold applied during HHpred analysis

Results section

Supplementary figure S2

Nice micrograph. However, it would be great if you could have used a less crowded micrograph for easy visualization and estimation of phages' dimensions.

L533-551: Needs to be un-bold.

Conclusion

 (L724-740): This section is not clear. Please make it short and straight to the point for easy understanding by the readers.

Author Response

Answers of authors to Reviewer 1

R1: It is a well done work that deserves to be published by this journal after the below minor issues have been rectified.

                We would like to thank the respected reviewer 1 for positive evaluation of our efforts and pointing out some issues that were in need of improvement and are now addressed in the second version of the manuscript.

R1: Parts that needs revision are:

Materials and Method Section

Please use RCF units (…xg) instead of RPM units

                We agree that indicating relative centrifugal force is a more appropriate way of ensuring replicability of the centrifugation conditions and compensate for RCF differences under same RPM due to potential differences in used rotor diameters. This has been overlooked by us initially and the units have now been changed from RPM to RCF in the revised version of the manuscript. Thank you for this suggestion.

R1: State the product catalogue number company name; for all the material (e.g media, enzymes and filters) used. (L100, L103, L106, L109-110, L125 )

                Thank you for this remark. Company name was, indeed, absent in some of the lines throughout the text (including line 100 for Proteinase K, line 101 for SDS etc.), this has now been addressed, and all materials should have the originating company mentioned now. Mentioned lines 106, 109-110 and 125, however, did include company names in the initial version of the manuscript. LB medium was prepared in our lab, the recipe is provided in the text, and its manufacturers of its constituent ingredients are now provided as well. As for the catalogue  numbers, although this can be done from documentations associated with the lab orders, we do not see how could their inclusion benefit the manuscript or their absence hamper reproducibility of the research presented in the manuscript. The main issue with the product catalogue numbers is that these numbers are subject to change, thus they may become irrelevant and puzzling for readers in the near future. As, for example, for line 125 (Custom primers ordered from Metabion), no catalogue number can be provided per se. However, If the editor agrees with the reviewer 1 that catalogue numbers are necessary, they can be included in the next round of revisions, where possible.

R1: As for machines like TEM: state their model/name, include the town or city and country where the equipment was made (L92, L95, L105, L112).

                We have tried to provide models/names for specific equipment used during this study in the first version of the manuscript, as well as indicate the corresponding manufacturers, which was the information directly available to us at the time. We have now also tried to provide manufacturer country for specific equipment used based on the company headquarter information available at their websites in the second version of the manuscript. However, we do not understand how could it matter in which town or city the particular piece of equipment was made for reproducibility of the research presented. The only way of obtaining that information we can think of would probably be directly contacting the manufacturers with equipment identifiers. However, we are not sure whether the replies of manufacturers could be received until our revisions are due. If the editor and reviewer 1 insist that the town or city where equipment was made is needed, we would like to ask for another round of revisions with a longer time for our replies.

 R1: L85:     …QHP IEX- State its full meaning for the readers.

QHP IEX stands for “Q Sepharose® High Performance ion exchange chromatography”. The QHP IEX acronym has now been removed in favor of full meaning, which is now presented in the second version of the manuscript. We are sorry for causing possible confusion with this acronym.

R1: L86-87:

  1. Which medium was used during phage concentration? ...and at what RCF (..xg) were samples spun?

By this point the fractions that went to concentration were in PBS buffer of ~7.3 pH. They were spun at 3214 RCF on Eppendorf 5810R centrifuge until the volume has reached ~0,5 mL (sadly, the exact time how long did it take is not mentioned in the lab notes).

  1. Alternatively you may cite the authors whose method was used.

Not really applicable as these conditions were chosen empirically based on the previous work with phages in our lab, concentrating phages exactly that or similar way is something we do routinely.

R1: L142: State an e-value threshold applied during HHpred analysis

                The default parameters were applied during the HHpred analysis (“E-value cutoff for MSA generation 1e-3”), the “default settings” is now mentioned in the text. We thank the Reviewer 1 for this remark, as it was also noted by us that we have mistakenly not mentioned that HHpred analysis was actually ran on four selected databases/sources: Protein Data Bank (PDB), Pfam, UniProt-SwissProt-viral70 and NCBI CD, instead of just the default PDB option, this is now stated in the second version of the manuscript. Typo was noted in BLASTp e-value threshold, the final value used was 1e-3, instead of 1e-4 (although it did not impact anything apart from having an additional hit to ORF #39 identifiable). This has also been addressed in the second version of the manuscript.

R1: Results section

Supplementary figure S2

Nice micrograph. However, it would be great if you could have used a less crowded micrograph for easy visualization and estimation of phages' dimensions.

                Sadly, we did not prepare any of the diluted samples for TEM at the time, which could result in a less crowded micrographs. This micrograph is a representative field of view of the sample that went to TEM. We are aware that this picture is not of the best quality, as it is not only crowded, but also contains defective Nocturne116 particles or their parts in addition to the intact virions, thus the reason of “hiding” it in the supplementary materials associated with the manuscript. As for phage particle dimensions, we have taken care of estimating the intact virion dimensions and present them in lines 198-200 of the both versions of the manuscript.
                Taking into account that the main focus of this manuscript was mostly on the genomic characterization of Nocturne116 (thus submission to Microorganisms’ “Bacteriophage Genomics” special issue), we have decided not to arrange an another round of TEM imaging as the morphological features of the phage in question are still discernible from the micrograph presented as Supplementary figure S2 and do not take an important place in this study.

R1: L533-551: Needs to be un-bold.

                We are sorry for this formatting mistake, this section has now been unbolded in the second version of the manuscript.

R1: Conclusion

 (L724-740): This section is not clear. Please make it short and straight to the point for easy  understanding by the readers.

We are sorry if the conclusion section seems confusing to Reviewer 1. We have to agree that, taking the concept of the manuscript into the account, the section with the results and discussion has proven to be massive, with different points being made and discussed throughout it. In lines 724-740, which represent the “Conclusions” section, we have tried our best to sum up the “Results and Discussion” in a concise manner. We have opted to include three paragraphs in this section, representing the main “take-home messages” signified by this research. This way, the first paragraph states the need for careful manual annotation of novel phages, the second paragraph states that even such an approach might be hindered by sufficient divergence of a novel phage from all the other currently known phages, and the third paragraph, basically, concludes on the uniqueness of Nocturne116 amongst Lactococcus phages.
                While above we have tried to provide our justification on the structure of our “Conclusions” section, we are open for specific suggestions on how to make it more easily readable (what should be excluded or included, as perceived by the respected Reviewer 1), while making sure it still conveys the study in a very concise matter.

As for the language throughout the manuscript – we have reread it once again and did spot some spelling/grammar/language mistakes, which are now also corrected to the best of our knowledge with the goal of improving the overall English. If the language quality is still perceived insufficient, and the necessary improvements are deemed out of typesetting scope that should take place in case of acceptance, we can probably try arranging a professional language editing prior to possible publishing.

Reviewer 2 Report

The manuscript is well written and the genome of a potential new lactococcus phage genus is characterized.

I totally agree with you about the need of an as you stated manual supervised annotation approach at least when introducing new Phage species/genera. I appreciate the effort you made to annotate the described orfs.

Unfortunately, I totally miss a short review of lactococcus phage diversity in the introduction. As you claim that Nocturne116 might be a new phage genus I would expect to find information of the current state of lactococcus phage taxonomy. I think that it should not be a surprise that there are currently 14 known Lactococcus phage genera (line 607), as there is plenty of literature describing the diversity and taxonomy of lactococcus phages. Though it seems that information about the phage genera is missing in public sequence repositories. As you correctly stated Lactococcus has the most publicly available corresponding phage genomes, this is due to the importance of Lactococcus in the dairy industry. Consequently, there is a lot of research analysing these phages and their taxonomy. Skunabirus (936 or sk1 phages) and Ceduovirus (or c2 phages) are the most relevant phages in dairies therefore the biggest groups.

In line 199, while describing the morphology of phage Nocturne116 consider to compare it to the next possible relative (Q54, also with a prolate capsid) as a prolate capsid is not the most common morphology within the Lactococcus phages.

Also, the TEM picture (supplementary_figure_S2) has a lot of defect phage particles with a lot of empty heads and lose tails making it hard to even find an intact phage particle. If possible change the picture.

Author Response

Answers of authors to Reviewer 2

R2: The manuscript is well written and the genome of a potential new lactococcus phage genus is characterized.

I totally agree with you about the need of an as you stated manual supervised annotation approach at least when introducing new Phage species/genera. I appreciate the effort you made to annotate the described orfs.

                We are sincerely happy that another scientist in the field (Reviewer 2) agrees on the existence of such a problem (supervised genome annotation being often being overlooked in novel phage characterization process/papers, which leads to sub-par phage genome annotation qualities in some cases) and has positively evaluated our efforts to show an elaborate way of trying to overcome it, that should nevertheless be easily adoptable for any novel phage being annotated using a description of Nocturne116 genome annotation in this manuscript as an example in a way. We also thank the respected Reviewer 2 for suggestions on improving parts of this manuscript.

R2: Unfortunately, I totally miss a short review of lactococcus phage diversity in the introduction. As you claim that Nocturne116 might be a new phage genus I would expect to find information of the current state of lactococcus phage taxonomy. I think that it should not be a surprise that there are currently 14 known Lactococcus phage genera (line 607), as there is plenty of literature describing the diversity and taxonomy of lactococcus phages. Though it seems that information about the phage genera is missing in public sequence repositories. As you correctly stated Lactococcus has the most publicly available corresponding phage genomes, this is due to the importance of Lactococcus in the dairy industry. Consequently, there is a lot of research analysing these phages and their taxonomy. Skunabirus (936 or sk1 phages) and Ceduovirus (or c2 phages) are the most relevant phages in dairies therefore the biggest groups.

                We are thankful to you for this special remark! We totally agree that Lactococcus phage diversity (not surprising, with Lactococcus-infecting phages being one of the most abundant by total number of completely sequenced phages infecting particular genera) is a worthwhile topic, probably worthy of a dedicated fully-fledged review summarizing all of the recent discoveries regarding phages of Lactococcus spp. in one place by one of the stalwarts of the Lactococcus phage research (which none of the authors is). After the relevant scientific literature search it seems that all of the qualitative reviews on Lactococcus phages we were able to find might as of now be considered dated in the light of the continuing Lactococcus phage research.  However, this manuscript “Genome characterization of Nocturne116, novel Lactococcus lactis-infecting phage isolated from moth” is already a rather massive one, with a strong focus on a particular phage, which is not actually similar to any other Lactococcus phages as shown here (save for a distant link to Q54), thus we have opted to make the introduction as short as possible, as it would be just impossible, in our opinion, to do overall Lactococcus phage diversity literature summary/review justice in the introduction within the scope of this paper without making the manuscript considerably lengthier.

Our lab is working with different phage-host pairs isolated from the variety of environmental sources using culture-based approaches, not focusing specifically on Lactococcus spp. and their phages, or any other particular hosts/phage groups. We actually focus on describing isolation and characterization of peculiar phages of any host we can retrieve from the environment with the goal of expanding the known phage diversity via such a “bizarre” approach. Although we acknowledge that, strictly speaking, the phage diversity expansion would be somewhat more readily done via meta- genomics/viromics approaches that do not require culturing, we believe that culture-based approaches open greater possibilities to gain further novel insights about most phages found (having an actual phage and its host in a lab, rather than only a contig representing phage genome or its part).

Having a vast array of host-phage objects in our lab (one of them being Lactococcus lactis isolate LNT culture and phage Nocturne116) and not having any prior focus on Lactococcus phages was the reason for our great surprise after having a deeper look into genomes/proteomes of Lactococcus phages that were unidentified at the genus level based on their genome submission taxonomy at the time of performing this study, as well-defined clusters corresponding to genera were readily identifiable. The official shifts in Lactococcus  infecting phage taxonomy were observed by us only during the study and not actually prior to it, thus, we have opted to include the mention of the current state of Lactococcus phage taxonomy in a dedicated section (3.3) of the “results and discussion”, which we named “The current state of Lactococcus phage genomic diversity” for this information to have a better connection to the results we have observed and highlight the significant lag identified before ICTV updates are incorporated to public biological sequence repositories (which has actually happened as of now (15.07.2021), but was not the case while the study was performed and submitted). Although now debatable, we still believe that the information on Lactococcus phage taxonomy is better suited to remain in this dedicated section of the results and discussion to preserve the original flow of the study presented.

                Although above we have tried to justify our view of why we do not find such changes necessary to benefit this paper, as, we believe, the requested information is there, but in another place. Should the reviewer 2 disagree with our point of view, we could try reworking parts of section 3.3 “The current state of Lactococcus phage genomic diversity” to make them appropriate for an introduction if reviewer 2 finds it better suitable there. Either, if the reviewer 2 insists and the editor supports this decision, we shall oblige, and could try to do our best to cover the recent Lactococcus phage research advancements in introduction in a reasonable time, which would add up some new paragraphs to it in the next round of revision, for which we would then like to ask for a lengthier due time, should this be the case.

In addition, although the taxonomy update in repositories does not alter the conclusions of this study in any way, a post-scriptum was now also added to the end of the results and discussion section in the second version of the manuscript, stating that the Lactococcus phage taxonomy has finally been updated in public sequence repositories.

Lines 741-745 now read:
 “P.S. While making revisions to the initial version of this manuscript during the review process, it was noted by us that on 15.07.2021 the genus level taxonomy associated with the Lactococcus phage complete genome submissions has finally been updated in public biological sequence repositories to conform with the current official ICTV taxonomy ratified in March 2021 after few month-long lag.”

R2: In line 199, while describing the morphology of phage Nocturne116 consider to compare it to the next possible relative (Q54, also with a prolate capsid) as a prolate capsid is not the most common morphology within the Lactococcus phages.

We are grateful to the reviewer 2 for this idea and see how this could benefit the manuscript. The line numbers have shifted after some language adjustments were made and concerns of the reviewers been addressed, but the “Overview of Nocturne116 genome” section does now include phage Q54 and Nocturne116 high-level similarity description, including particle dimension comparisons.
The lines 222-228 were added in the second version of the manuscript, these read:
“It is worth noting that, despite high overall nucleotide sequence diverge to Noc-turne116, Q54 [31], the closest relative of Nocturne116, boasts not only similar genome size, organization, and ORF count (26537 bp with predicted 47 ORFs for Q54 versus 25554 bp with 52 predicted ORFs for Nocturne116), they both also show exactly the same 3’ co-hesive genome termini and similar virion morphology (prolate capsids of 56x43 nm for Q54 and ~59x40 nm for Nocturne116 with 109 nm versus ~112 nm long non-contractile tails, respectively).”

R2: Also, the TEM picture (supplementary_figure_S2) has a lot of defect phage particles with a lot of empty heads and lose tails making it hard to even find an intact phage particle. If possible change the picture.

The Nocturne116 micrograph presented as supplementary figure S2 has also raised concerns of Reviewer 1.

                We are aware that this picture is not of the best quality, as it is not only contains defective Nocturne116 particles or their parts in addition to the intact virions, but also is somewhat too crowded. Sadly, we did not prepare any of the dilutions of this sample for TEM at the time, which could result in a less crowded micrographs, but would probably still contain defect particles or their parts. This micrograph is a representative field of view of the sample that went to TEM, and other samples were not prepared and pictured back then. Although another sample could probably be prepared from the Nocturne116 stock  in storage at our lab, using less experimental interventions that could lead to virion damage, thus, more “intactness” of the particles observable, we hardly see justification of obtaining a more qualitative picture as the morphological features of the phage in question are still discernible from the micrograph presented as Supplementary figure S2 and morphology of Nocturne116 does not play an important role in this largely genomic study (that was specifically tailored for submission to Microorganisms’ “Bacteriophage Genomics” special issue). The somewhat lackluster quality of the micrograph is actually the reason we have “hid” it in the supplementary materials associated with the manuscript and have taken care of estimating the intact virion dimensions to present them to readers in lines 198-200 ourselves.

As for the language throughout the manuscript – we have reread it once again and did spot some spelling/grammar/language mistakes, which are now corrected to the best of our knowledge with the goal of improving the overall English language. If the language quality is still perceived insufficient, and the necessary improvements are deemed out of typesetting scope that should take place in case of acceptance, we can probably try arranging a professional language editing prior to possible publishing.